# Neurocognitive Deficits in First-Episode and Chronic Psychotic Disorders: A Systematic Review from 2009 to 2022

**DOI:** 10.3390/brainsci13020299

**Published:** 2023-02-10

**Authors:** Nadja Tschentscher, Christian F. J. Woll, Julia C. Tafelmaier, Dominik Kriesche, Julia C. Bucher, Rolf R. Engel, Susanne Karch

**Affiliations:** 1Section of Clinical Psychology and Psychophysiology, Department of Psychiatry and Psychotherapy, LMU Hospital Munich, Nußbaumstr. 7, 80336 Munich, Germany; 2Section of Clinical Psychology of Children and Adolescents, Department of Psychology and Educational Sciences, Ludwig Maximilian University of Munich, Leopoldstr. 13, 80802 Munich, Germany

**Keywords:** schizophrenia, psychotic disorders, neuropsychology, cognition, deficits, impairment

## Abstract

Cognitive impairment in patients suffering from schizophrenia spectrum disorders has been discussed as a strong predictor for multiple disease outcome variables, such as response to psychotherapy, stable relationships, employment, and longevity. However, the consistency and severity of cognitive deficits across multiple domains in individuals with first-episode and chronic psychotic disorders is still undetermined. We provide a comprehensive overview of primary research from the years 2009 to 2022. Based on a Cochrane risk assessment, a systematic synthesis of 51 out of 3669 original studies was performed. Impairment of cognitive functioning in patients diagnosed with first-episode psychotic disorders compared with healthy controls was predicted to occur in all assessed cognitive domains. Few overall changes were predicted for chronically affected patients relative to those in the first-episode stage, in line with previous longitudinal studies. Our research outcomes support the hypothesis of a global decrease in cognitive functioning in patients diagnosed with psychotic disorders, i.e., the occurrence of cognitive deficits in multiple cognitive domains including executive functioning, memory, working memory, psychomotor speed, and attention. Only mild increases in the frequency of cognitive impairment across studies were observed at the chronically affected stage relative to the first-episode stage. Our results confirm and extend the outcomes from prior reviews and meta-analyses. Recommendations for psychotherapeutic interventions are provided, considering the broad cognitive impairment already observed at the stage of the first episode. Based on the risk of bias assessment, we also make specific suggestions concerning the quality of future original studies.

## 1. Introduction

A global cognitive impairment in patients suffering from, e.g., schizophrenia spectrum disorders, persistent delusional disorders, or acute psychotic disorders, here all summarized under the term “psychotic disorders”, has been widely observed and is commonly accepted to be one of the main characteristics of these mental disorders [1,2,3]. In comparison to healthy controls, deficits have been observed in a wide range of cognitive domains, including attention, processing speed, memory, executive functioning, problem solving, working memory, and language [4,5]. Cognitive impairment in psychotic disorders has been considered as highly predictive for patients’ treatment outcomes, everyday functioning, and perceived quality of life [6,7]. Thus, it is of high therapeutic relevance to precisely specify the cognitive domains and severity of deficits in psychotic disorders, as well as the interaction with illness-related factors, such as the frequently observed progression of the disease from the first episode to the chronic stage.

It has been suggested that schizophrenia might be best understood as a neurodevelopmental disorder [8,9,10], with an early onset of premorbid cognitive impairment during childhood years before the first psychotic episode [11], which is followed by a progressive increase in cognitive deficits during adolescence until the occurrence of the first psychotic episode [12]. However, little is known about the trajectory of cognitive impairment following the transition from the first episode to the chronic stage. Additionally, while the number of primary studies on psychotic disorders has continually increased over the past decades [13], there have been only a few attempts to synthesize these research outcomes [4,14,15]. This might be partly due to the heterogeneity of the published studies concerning design, data quality measures, sample selection, and focus of research [15]. Fioravanti and colleagues [14] list a number of common methodological problems including small sample sizes, clinical diversity of patient groups, and divergent statistical approaches, resulting in an estimated heterogeneity across studies of about 80 percent.

In a recent meta-analysis on neurocognitive deficits at the high-risk stage for developing a psychotic disorder, an in-depth quality assessment was performed, and results were drawn from a well-characterized sample of 78 studies [4]. The main finding was a widespread impairment of cognitive functioning already in individuals at the high-risk stage, as compared to healthy controls. Furthermore, the severity of cognitive impairment within specific cognitive domains was predictive for transitioning to the first episode of a psychotic disorder. Here, we extend this work by focusing on cognitive deficits at the first-episode stage, and on those after transitioning to the chronic stage. Previous meta-analyses on first-episode and chronic psychotic disorders have covered primary studies from the years 2005 to 2007 [2], and from the years 2005 to 2010 [14], respectively. Here, we follow this line of research by including the most recent and methodologically advanced primary studies from the years 2009 to 2022, to provide a smooth continuation of previous work.

To allow accurate reproducibility of previous findings, we used the previously applied search terms, but also extended the list of search terms and databases, as defined by Catalan et al. (2015), to capture research on the progressed stages of illness. So far, there is no comprehensive evidence on whether the commonly reported steep decline in cognitive abilities at the first-episode stage of psychotic disorders is followed by the progression or remission of deficits within specific cognitive domains. A recent narrative review has highlighted the conflicting evidence across longitudinal studies [16]: while there is, overall, a moderate tendency towards the persistence of some cognitive impairment in *most* patients, longitudinal studies show, for example, inconsistent results concerning the changes in fluid versus crystalized intelligence components over time. Moreover, heterogeneous evidence exists on whether there is a specific subset of patients showing progressive deterioration of cognitive functioning in later life [17], which has been interpreted as an accelerated aging process [18,19].

We carefully selected studies with comparable study designs, as quantified by our in-depth risk-of-bias assessment. A comprehensive neurocognitive classification scheme was used, which included the following cognitive domains reported by a recent well-designed meta-analysis on neurocognitive deficits in individuals at the clinically high-risk and first-episode stages [4]: attention and vigilance, executive functioning, language, social cognition, verbal cognition, visuomotor processing, visual—spatial cognition, and working memory. As global decline in cognitive functioning can be assumed in patients diagnosed with a psychotic disorder [3], with deficits in cognitive functioning observed to be most severe in schizophrenia but also present in groups of patients diagnosed with other psychotic disorders [5], we defined a (1) *global deficit hypothesis*, which is closely linked with the neurodevelopmental model of schizophrenia [10]. Cognitive deficits in patients diagnosed with psychotic disorders were predicted for all reported cognitive domains, compared with healthy matched controls [2,4,20]. Concerning the trajectory of cognitive impairment, previous studies suggest a substantial decline in cognitive functioning before or at the stage of the first episode, and a persistence or increase in impairment at the chronic stage of illness [21], as reflected in our (2) *longitudinal stability hypothesis*. Cognitive deficits were expected to be frequent both at the first-episode stage, as well as at the chronic stage of illness [17], while specific cognitive domains might show more frequent impairment at the chronic stage of illness, in line with the assumption of an accelerated aging process [18,19]. In exploratory analyses, we also included studies on patients with a high risk for psychotic disorders to evaluate whether cognitive profiles of first-episode and chronically affected patients resemble those at the high-risk stage [5].

## 2. Methods

### 2.1. Pre-Registration

In order to guarantee the transparency and reproducibility of our research work, we pre-registered the introduction and method sections with the Open Science Framework (see https://osf.io/nyk4s, accessed on 24 October 2019), adhering to the PRISMA 2020 reporting guidelines [22]. Additionally, we uploaded our search strategy and literature hits from the applied databases in common file formats, so that they can be accessed publicly, hereby allowing other researchers to transparently back-trace and re-analyze our data (see https://osf.io/tngfc/, accessed on 1 February 2023). Changes to the pre-registration have been made in the current systematic review concerning the period of included primary research: due to delays in the workflow, we extended our search period to January 2022, while we originally planned to consider literature hits from the period 2009 to 2019. The current changes are addressed in the Open Science Framework, and the literature hits from the years 2009 to 2022 are uploaded accordingly.

### 2.2. Inclusion Criteria

We included primary, peer-reviewed studies published in English by international, well-established journals. We defined the research question according to the P(I)COS statement in terms of population (P), comparators (C), outcomes (O), and study design (S). Only the data of patients and matched control groups from longitudinal or intervention studies (I) on therapeutic or medication treatments were included from the pre-intervention baseline assessment. Concerning the population (P), all included studies examined adults and teenagers with an age of at least 16 years who met the criteria for the primary diagnosis of a psychotic disorder according to international diagnostic manuals (for example, ICD-10, DSM-IV, and DSM-5). We excluded studies on patients suffering from schizoaffective illness. The control for comorbid illnesses was addressed in our risk-of-bias assessment. We excluded all review articles, meta-analyses, studies on animals, and studies focusing on comorbid psychological diseases (for example, depressive disorder or substance abuse disorder). Studies lacking a matched control group of healthy participants as comparators (C) were excluded as well. Furthermore, studies with a primary focus on the relatives of patients and those only focusing on patients at the remitted stage were excluded. All included studies had performed an assessment of neurocognitive functioning by using reliable, valid, and objective neuropsychological instruments as outcomes (O). In order to focus on current scientific production, we only included studies published between 1 January 2009 and 31 January 2022. Concerning the study design (S), only studies allowing a between-group comparison between a patient group and a healthy control group were included. All longitudinal or intervention studies without a healthy control group at baseline level were excluded.

### 2.3. Data Sources and Search Terms

The online databases PsycINFO, Scopus, and PubMed were used for the literature search. The following search term was selected based on the careful assessment of previous review articles, meta-analyses, and titles of suitable primary studies in the field, and was applied to all selected databases: “(schizophren* OR psychotic OR psychosis) AND (cogniti* OR neuropsycholo*) AND (impairment* OR function* OR deficit*)”.

“(schizophren* OR psychotic OR psychosis) Neurocognitive Deficits In Schizophrenia 5 AND (cogniti* OR neuropsycholo*) AND (impairment* OR function* OR deficit*)”.

Additionally, the option to search for “linked full texts” was deactivated for PsychINFO, publications were restricted to the years from 2009 to 2022, the publication type was set to “peer reviewed journal”, and the box “English” was ticked. For PubMed, we chose “13 years” as the filter for the publication dates. For Scopus, the search settings were restricted to the years from 2009 to 2022, the document type was set to “article”, and the access type to “All”. Using the described search settings, we received 984 initial hits in PsychINFO, 1434 in PubMed, and 1251 in Scopus.

### 2.4. Study Selection and Data Extraction

The study selection and data extraction were carried out by three different members of the research team. Excluded studies were assigned to different categories, depending on the reason of exclusion (see Figure 1 for outline of the study selection process). Studies that seemed to be relevant were downloaded in a RIS format and saved in Citavi. The first author (N.T.) double-checked the excluded and included studies. From included studies, the following data were extracted: author names, publication year, date and place of the study, diagnosis, age of patients and healthy controls, applied neuropsychological tests, and the test outcomes.

### 2.5. Risk of Bias Assessment

All included studies were assessed for their risk of bias by three authors (N. T., J. T., and J. B.). We evaluated the risk of bias using the relevant domains of the Cochrane risk of bias tool for randomized controlled trials [23], and of the risk of bias in non-randomized studies of interventions (ROBINS-I) tool [24], i.e., “selection bias”, “blinding of patients”, “detection bias” (i.e., blinding of assessors), “complete vs. incomplete data reporting”, and “free of selective result reporting”. Additionally, we assessed whether a clear and thorough diagnostic procedure was applied. A summary of the risk of bias assessment for each included study is presented in Table 1. In line with the Cochrane risk of bias tool [23], studies were either categorized as carrying a low risk (+) if the criteria were met concerning all assessed domains, as carrying an unclear risk (?) if the evaluation of at least one domain provided an unclear outcome, or as carrying a high risk (-) if the criteria of at least one assessed domain were not met. We refrained from using a finer-grained quality scale, as the resulting scores are not an appropriate way to assess clinical trials because they, for example, assign weights to studies concerning their level of risk in ways that are difficult to justify [23]. Detailed descriptions concerning each risk of bias category for each of the included studies are provided in an Excel coding sheet in the OSF (see https://osf.io/nyk4s, accessed on 24 October 2019).

### 2.6. Synthesis of Study Outcomes

Concerning our specific hypotheses on the cognitive impairment of patients with first-episode and chronic psychotic disorders, we extracted and summarized patient groups belonging to either one of these categories from primary studies. For exploratory reasons, we also extracted and summarized the data of study groups carrying a high risk for psychosis and those of the siblings of first-episode patients. We primarily focused on the cognitive domains of the MATRICS Consensus Cognitive Battery (MCCB) [25], based on which we defined eight domains and some subdomains: attention and vigilance, executive functioning (including the subdomains abstraction, cognitive flexibility, inhibition, planning, and reasoning and problem solving), language (including the subdomains comprehension, naming, verbal fluency, and vocabulary), social cognition, verbal cognition (including the subdomains verbal learning and verbal memory), visuomotor processing, visual–spatial cognition (including the subdomains visual analysis and construction, and visual learning and memory), and working memory. The relation of specific cognitive domains and psychometric tests is presented in Table 2. Only statistically significant results of reliable, valid, and standardized neuropsychological tests were extracted. Descriptive evaluations (for example, “better” and “higher scores”) were not considered for the systematic review. In our synthesis of study outcomes, we focused on the frequency of significant differences (patients versus healthy controls) across studies for a specific clinical subgroup and cognitive domain. The frequency of cognitive impairment in a particular domain was calculated across studies using percent quantiles if each cognitive domain was investigated by at least 3 studies. Quantiles of the frequency of cognitive impairment across studies were calculated for the respective patient groups by dividing the number of studies reporting deficits in a particular domain by the total number of studies that assessed this domain.

## 3. Results

### 3.1. Included Studies and Clinical Subgroups

A total of 3669 studies were screened for eligibility, of which 51 studies were included. A flowchart addressing the number of excluded studies, the reasons for their exclusion, and the search and screening process is presented in Figure 1. The cognitive performance scores of the following clinical subgroups, as defined by the primary studies, have been included in this systematic review (for main results, see Table 3):(1)chronic schizophrenia (CHS);(2)deficit chronic schizophrenia with stable negative symptoms (CHS-D);(3)non-deficit chronic schizophrenia (CHS-ND);(4)chronic schizophrenia treatment non-responders (CHS-NR);(5)chronic schizophrenia treatment partial responders (CHS-PR);(6)chronic schizophrenia without motor retardation (CHS-NMR);(7)chronic schizophrenia with motor retardation (CHS-MR);(8)chronic schizophrenia without oxidative stress (CHS-NOS);(9)chronic schizophrenia with oxidative stress (CHS-OS);(10)high risk of psychotic disorders (HRP);(11)early stages of high risk for psychotic disorders (HRP-early);(12)late stages of high risk of psychotic disorders (HRP-late);(13)first-episode psychotic disorder (FEP);(14)first-episode schizophrenia (FES);(15)non-remitted first-episode schizophrenia (FES-NR);(16)remitted first-episode schizophrenia (FES-R);(17)schizophrenic patients (SZ).

Data of patient groups were analyzed relative to those of a matched healthy control group (HC). For our specific hypotheses, we focused here on the main categories “first-episode” and “chronic stage” of psychosis, under which all respective subgroups above were summarized. For exploratory reasons, we also included the categories “high-risk” and “siblings of first-episode patients”. The data of the siblings of first-episode patients (S-FEP) were considered in exploratory analyses.

### 3.2. Risk of Bias Assessment

Based on the modified Cochrane risk of bias tool, 15 studies carried a high risk of bias, and 36 studies carried an unclear risk of bias. None of the studies had a low risk of bias (see Table 1). Out of the high-risk studies, a risk of bias was detected in one up to three different categories per study: 10 studies showed methodological deficits in the sample selection process, 7 studies did not report on the blinding of patients, 2 studies showed an incomplete reporting of data, and 1 study showed a selective reporting of research outcomes. The remaining studies showed an unclear risk of bias, mainly due to a lack of reporting on blinding measurements, as well as on the diagnostic process. Overall, our set of included studies carries an unclear risk of bias. Therefore, the results and conclusions of this review must be interpreted with caution.

### 3.3. Cognitive Performance of Patient Groups vs. Healthy Control Groups

The study group characteristics, the applied cognitive tests, and the main research outcomes of all studies are presented in Table 3. For additional information on study group characteristics, please see Appendix A. A summary of results is presented in Table 4. Quantiles of the frequency of cognitive impairment across studies were been calculated for the groups of first-episode and chronically affected patients, respectively, by dividing the number of studies reporting deficits in a particular domain by the total number of studies that assessed this domain. For studies with multiple subgroups (for example, chronic schizophrenia treatment non-responders and chronic schizophrenia treatment partial responders) we only included outcomes of a particular domain if they did not show contradicting results (i.e., if tests across subgroups were either consistently significant or did not show any significant differences). There are not sufficient data on the group of high-risk patients to perform this summary assessment: only 6 out of 51 studies included participants considered at high risk of developing psychotic disorders. This sample is, therefore, not included in the results summary.

At the stage of the first psychotic episode, strong evidence for cognitive deficits in psychotic patients was observed for most of the assessed cognitive domains, when compared with healthy matched controls. Cognitive deficits were reported in 100 percent of studies assessing the following domains (the number of studies that assessed the respective domain is indicated in brackets): abstraction (N = 3 studies), cognitive flexibility (N = 10), inhibition (N = 8), naming (N = 3), planning (N = 3), reasoning and problem solving (N = 10), verbal fluency (N = 18), verbal memory (N = 3), and visuomotor processing (N = 22). More than 75 percent of studies reported cognitive deficits in the following domains: attention and vigilance (N = 20), social cognition (N = 9), verbal learning (N = 20), visual learning and memory (N = 13), and working memory (N = 18). Weaker evidence for cognitive impairment was only found for the domain of “visual analysis and construction” (N = 6), for which 50–75 percent of studies reported deficits, and for the domain “vocabulary” (N = 4), for which 25–50 percent of studies reported deficits.

At the chronic stage of psychotic disorders, cognitive impairment was observed in 100 percent of studies assessing the following domains (the number of studies that assessed the respective domain is indicated in brackets): cognitive flexibility (N = 9), inhibition (N = 9), verbal fluency (N = 22), verbal memory (N = 6), visual analysis and construction (N = 4), and visuomotor processing (N = 25). More than 75 percent of studies reported cognitive deficits in the following domains: attention and vigilance (N = 20), planning (N = 7), reasoning and problem solving (N = 16), social cognition (N = 11), verbal learning (N = 19), visual learning and memory (N = 14), and working memory (N = 24). Weaker evidence for cognitive impairment was only found for the domain of “vocabulary” (N = 3), for which 50–75 percent of studies reported deficits. We also assessed the putative impact of medication on the degree of reported cognitive impairment across studies on patients at the chronic stage of illness. In 38 percent of these studies, antipsychotic treatment was applied to chronically ill patients. A total of 14 percent of studies reported a mixed sample of patients, in which the majority but not all patients received antipsychotic medication, while 7 percent of studies reported that no medication was given to patients. Most studies, i.e., 41 percent, did not provide sufficient information on medication. However, across the studies that reported on medication, there was no difference concerning the degree of cognitive impairment in chronically ill patients depending on the antipsychotic treatment.

Considering differences in the risk of bias across studies (Table 1), we also calculated the frequency of cognitive impairment in a particular domain excluding studies that carried a high risk of bias. For the remaining N = 17 studies at the first-episode stage and N = 18 studies at the chronic stage, we summarized results using percent quantiles if each cognitive domain was investigated by at least three studies. No differences in the overall frequency of cognitive impairment across domains were observed relative to calculations including all studies (Table 4). However, specific cognitive domains, e.g., vocabulary at the chronic stage, could not be assessed due to an insufficient number of studies.

While the strength of the evidence in support of cognitive impairment in a specific domain may vary depending on the indicated number of studies that assessed this domain, the results support the *global deficit hypothesis* overall, which predicts an extensively reduced cognitive performance in multiple areas already at the stage of the first psychotic episode. Furthermore, in line with the *longitudinal stability hypothesis*, our data showed a similar pattern of cognitive deficits in patients at the first-episode stage and at the chronic stage of psychotic disorders. More frequent impairment at the chronic stage, relative to the first-episode stage, was only observed across studies for the domains “visual analysis and construction” and “vocabulary”.

### 3.4. Cognitive Performance of First-Episode vs. Chronically Affected Patients

To further explore the longitudinal stability of cognitive impairment in psychotic disorders, we summarized results of studies that directly compared groups of patients at the first-episode and the chronic stage (Table 5). For the majority of the evaluated subdomains, no significant differences between first-episode and chronic-stage patients were observed, while both patient groups showed a significant impairment relative to a healthy control group in a number of cognitive domains (N = 34 significant subtests). The second most frequent contrast showed the strongest impairment in the chronically affected group, followed by the group of first-episode patients (N = 14 significant subtests). Thus, these data may support the *longitudinal stability hypothesis* of cognitive deficits at the chronic stage of illness, while little evidence is provided for a remission of cognitive impairment in psychotic disorders over time.

### 3.5. Cognitive Profiles of High-Risk Patients

Exploratory analyses of six studies on patients at the high-risk stage revealed deficits in multiple cognitive domains when compared with healthy matched controls. However, each of these studies also found a range of cognitive domains to be unimpaired for high-risk patients. Thus, there was a substantial heterogeneity of results across studies concerning the impairment of specific cognitive domains. For example, working memory was found to be impaired in two studies [38,57], while the other three studies reported no significant difference between patients and healthy controls [40,48,54]. Attention and vigilance was found to be impaired in three studies [40,47,57], while two other studies observed no impairment [38,48]. Overall, cognitive deficits of patients at the high-risk stage do exist in several domains. However, substantial variations in deficits can be found across studies, suggesting there might not be a specific profile of impairment at the high-risk stage.

### 3.6. Siblings of First-Episode Patients

The evaluation of two studies that included the siblings of first-episode patients revealed heterogeneous outcomes: while one study reported overall milder levels of cognitive impairment in siblings [35], the other study found siblings to be equally impaired as first-episode patients for the majority of the assessed cognitive subdomains [77].

## 4. Discussion

This systematic review provides a comprehensive overview of primary research on cognitive deficits in psychotic disorders from the years 2009 to 2022. We observed a consistent and broad decrease in cognitive functioning in patients diagnosed with psychotic disorders. The affected cognitive domains included, for example, executive functioning, memory, working memory, psychomotor speed, and attention. Cognitive profiles of patients diagnosed with a first psychotic episode highly resembled those of patients at the chronic stage of illness. Thus, across studies, most cognitive domains were found to already be frequently affected at the first-episode stage. Furthermore, our systematic review provides evidence for the presence of cognitive deficits at chronic stages of illness. This supports both our prediction that global deficits may already be present at the first-episode stage and our hypothesis on the longitudinal stability of cognitive deficits in patients diagnosed with psychotic disorders. While recent systematic reviews and meta-analyses mainly focused on the high-risk stage of psychotic disorders and the transitioning to the first-episode stage [4,15], this is, to our knowledge, the first systematic review evaluating the frequency of cognitive deficits at the first-episode and chronic stages of illness based on primary studies from the past ten years. We performed a thorough quality assessment for all included studies, and used a comprehensive neurocognitive classification scheme previously used for individuals at the clinically high risk and first-episode stages [4].

### 4.1. The Global Deficit Hypothesis

Our main finding of global cognitive deficits in patients diagnosed with psychotic disorders has been supported by a number of previous meta-analyses [4,14,78] and reviews [1,2,79,80]. While there is yet no singular factor defined as causing these deficits, supporters of a systemic hypothesis on cognitive impairment [81] have suggested that multiple neurobiological conditions may account for cognitive impairment in psychotic disorders, such as, for example, broad gray and white matter irregularities and abnormalities in glutamate and g-aminobutyric acid neurotransmission. Based on the current findings, there is no specific cognitive profile concerning the frequency of deficits across domains. However, a global loss of functioning in almost every cognitive domain might be a fingerprint of psychotic disorders, as well as a marker of distinction to the cognitive profiles of patients diagnosed with other psychiatric conditions. For example, it is well known that major depression [82] and bipolar affective disorders [83] are accompanied by impairment of cognitive functioning. However, in contrast to psychotic disorders, a typical pattern of impaired cognitive domains has been reported across primary studies: while processing speed and memory functions have been frequently observed to be reduced in affective disorders, executive functions and verbal fluency are mostly unimpaired [84]. Furthermore, cognitive deficits were observed to be milder in remitted depressed patients than in those suffering from an acute episode of depression. This suggests a demission of cognitive impairment in affective disorders over time, in contrast to the suggested persistence of cognitive deficits in patients diagnosed with psychotic disorders [85,86].

### 4.2. The Longitudinal Stability Hypothesis

This systematic review supports the claim that global cognitive impairment already emerges at the first-episode stage of psychotic disorders. We also provide comprehensive evidence in support of the longitudinal stability of cognitive deficits, i.e., that cognitive impairment was observed with a similar frequency across studies in both first-episode and chronically affected patients. Both the early onset of severe cognitive impairment, as well as its longitudinal stability, has been reported by previous meta-analyses covering primary studies from the years 2005 to 2007 [2], and from the years 2005 to 2010 [14]. Here, we show similar findings based on primary studies published in the years 2009 to 2022. Our results are further supported by longitudinal studies on cognitive impairment in psychotic disorders cf. [5,17]; most of these longitudinal studies suggest a stable cognitive profile from the first psychotic episode onwards. For example, only ten percent of patients showed a further decline in or a remission of impairment of executive function and attention measures over the course of the following two years after the first episode [87]. Additionally, the monitoring of schizophrenia spectrum patients over a ten-year period showed little changes in their cognitive profiles, even if an improvement of psychiatric symptoms occurred [85,86,88]. While our outcomes support these previous findings of the longitudinal stability of deficits overall, it is important to note that we did observe mild differences in the frequency of cognitive deficits between first-episode and chronic stage patients. At the chronic stage, visual analysis and construction, as well as vocabulary, were observed to be more frequently impaired, while planning as well as reasoning and problem solving were less frequently impaired, relative to the first-episode patients. This finding relates to previous assumptions that cognitive dysfunction in psychotic disorders could be partly explained by neurodevelopmental pathologies in early life [8,9,10], as well as by abnormal processes in later life, such as accelerated brain aging [18,19]. Specifically, this might explain why cognitive functions most closely associated with neurodevelopmental pathologies in adolescence, such as executive functions and problem solving, show the strongest impairment around the first episode, and might stabilize over the course of illness. Conversely, cognitive functions associated with crystallized intelligence, such as vocabulary, are thought to continuously evolve through adulthood, and might, therefore, show a further decline over the course of illness [89].

### 4.3. Limitations and Recommendations to Lower the Risk of Bias in Future Studies

The results of this systematic review provide evidence on the frequency of cognitive deficits in groups of patients diagnosed with psychotic disorders, relative to respective healthy control groups. However, we did not calculate effect sizes across study outcomes, meaning that information on the strength of cognitive impairment is lacking. Thus, the results of this review do not inform on the relative changes in cognitive functioning over time, i.e., whether milder or more severe impairment was observed in first-episode or chronically affected patients. They also do not inform on longitudinal changes within individual subjects, but rather focus on the cross-sectional frequency of impairment in groups of patients at different stages of illness.

Furthermore, the results of this systematic review should be interpreted with caution considering that all 51 included studies carried either an unclear risk of bias, or even a high risk of bias (see Table 1). Most studies with a high risk of bias showed limitations in the selection process concerning the characterization of clinical and healthy control groups (i.e., no matching of groups regarding premorbid intelligence of patients, age, or gender). Additionally, most studies did not assess patients for comorbid psychiatric or neurological illnesses, and did not execute a blinding procedure of assessors and/or study subjects. In our study sample, there was also heterogeneity concerning the specific diagnosis of psychotic disorders: all studies at the chronic stage of illness included schizophrenic patients only, while 7 out of 23 studies at the first-episode stage also included patients with other psychotic disorders. For future studies, we recommend conducting more specific analyses and outcome reporting for each of the diagnoses, e.g., schizophrenia spectrum disorders, persistent delusional disorders, or acute psychotic disorders. Furthermore, to improve the synthesis of research outcomes across studies, a reduction in the heterogeneity of the assessed cognitive domains and applied psychometric tests is highly recommended. In Table 6, we provide six practical recommendations for the design of future primary studies to lower the risk of bias and to improve the synthesis and replicability of research outcomes on neurocognitive deficits in psychotic disorders.

### 4.4. Recommendations for the Clinical Treatment of Patients with Psychotic Disorders

In this systematic review, we observed substantial cognitive impairment already at the stage of the first psychotic episode. Thus, we recommend a stronger incorporation of an early, standardized cognitive assessment in the clinical treatment of patients with psychotic disorders. A first cognitive assessment should be performed before the beginning of any therapeutic intervention, and follow-up assessments are recommended in intervals of about 6–12 months. For the clinical setting, regular cognitive assessments may help to predict moderating factors of the treatment, such as medication adherence, as well as the need for supplementary cognitive training [90,91]. Over the course of illness, cognitive assessment may inform, amongst motivational and social factors, about perceived life quality and the development of comorbid illnesses, such as depression [92]. As yet, the overall impact of cognitive enhancers, such as pharmacological agents targeting different neurotransmitter systems, was reported to be rather low [93]. Thus, the precise assessment of cognitive performance levels in combination with the application of specific cognitive compensatory approaches [94,95], such as internal self-management strategies and changes to the external environment, may help patients to adjust their life and work environments according to their specific needs.

In Table 7, we list recommendations for a stronger incorporation of neurocognitive performance factors into the planning and execution of clinical treatments for patients diagnosed with psychotic disorders.

## 5. Conclusions

In this systematic review, we observed severe cognitive impairment in both patients diagnosed with first-episode psychotic disorders and patients at the chronic stage of illness. The cognitive deficits included multiple domains, such as attention and vigilance, reasoning and problem solving, planning, inhibition, cognitive flexibility, learning and memory, verbal fluency, and working memory. More severe impairment at the chronic stage was only observed for visual analysis, construction, and vocabulary, and we did not find evidence for a remission of cognitive deficits at the chronic stage of illness. Based on our outcomes, we recommend standardized cognitive assessments before any therapeutic intervention, as well as on a regular basis over the course of clinical treatment. Cognitive assessment outcomes are essential for individualizing psychotherapeutic interventions, for predicting medication adherence, and for deciding on supplementary cognitive training, e.g., involving cognitive compensatory strategies.

## Figures and Tables

**Figure 1 brainsci-13-00299-f001:**
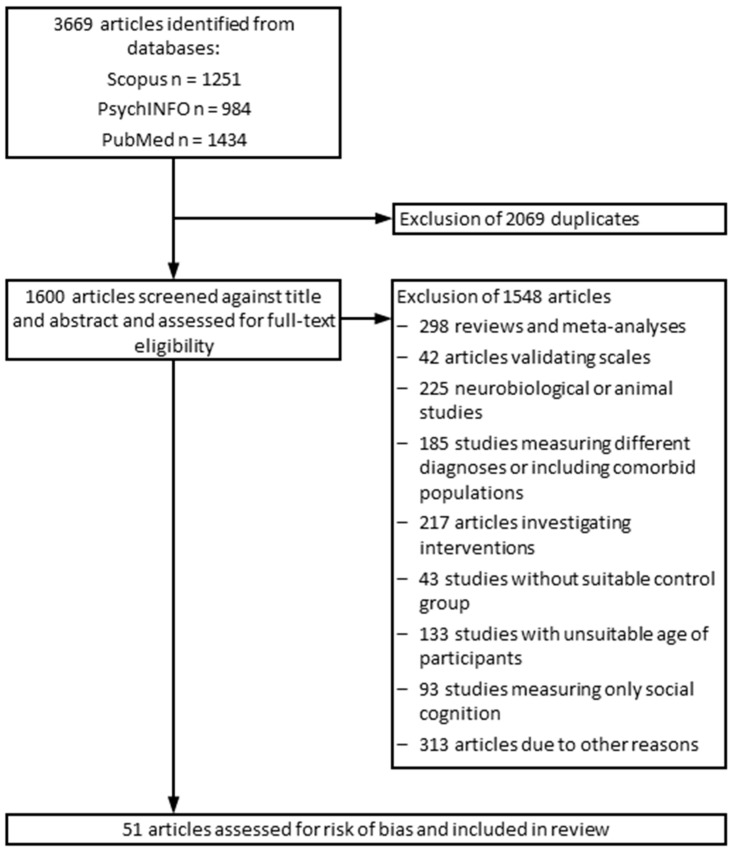
Flowchart outlining the study selection process.

**Table 1 brainsci-13-00299-t001:** Risk of bias assessment according to the Cochrane risk of bias tool [23]. Summary assessment: if the criteria were met concerning all assessed domains, the study was categorized as carrying a low risk (+). If the evaluation of at least one domain provided an unclear outcome, the study was categorized as carrying an unclear risk (?). If the criteria of at least one of the assessed domains were not met, the study was categorized as carrying a high risk (−).

First Author, Year	Selection Bias	Clear Diagnostics	Blinding (Patients)	Detection Bias	Incomplete Outcome Data Addressed	Free of Selective Reporting	Summary Assessment
Al-Dujaili [2021]	−	+	?	+	?	+	**−**
Al-Hakeim [2020]	−	+	?	+	+	+	**−**
Beck [2020]	+	?	?	?	+	+	**?**
Bliksted [2014]	+	?	?	?	+	+	**?**
Bosnjak Kuharic [2021]	−	?	−	?	?	+	**−**
Chattopadhyay [2020]	−	+	?	?	+	+	**−**
Chen [2019]	?	?	?	?	+	+	**?**
Chen 2021	−	+	?	?	+	+	**−**
Correa-Ghisays [2021]	−	?	?	−	+	+	**−**
Cuesta [2018]	?	?	?	+	+	+	**?**
Da Motta [2021]	−	?	−	+	+	+	**−**
De la Torre [2021]	+	?	?	?	+	+	**?**
Eisenacher [2018]	+	?	?	?	+	+	**?**
Ferretjans [2021]	−	?	?	?	+	+	**−**
Frommann [2011]	+	?	?	?	+	+	**?**
Giordano [2021]	?	?	−	?	+	+	**−**
Guo [2014]	+	?	?	?	+	+	**?**
Hájková [2021]	?	+	?	?	+	+	**?**
He [2013]	+	?	?	?	+	+	**?**
Konstantakopoulos [2020]	+	?	?	?	+	+	**?**
Koshiyama [2021]	−	?	?	?	−	+	**−**
Li [2018]	?	?	?	?	+	+	**?**
Liu [2019]	+	?	?	?	+	+	**?**
Liu [2021]	+	?	?	?	?	+	**?**
Maes, Sirivichayakul, Kanchanatawan et al. [2020]	?	+	?	+	+	+	**?**
Maes, Sirivichayakul, Matsumoto et al. [2020]	?	+	?	?	+	+	**?**
Maes [2021]	+	+	?	?	+	+	**?**
Mançe ÇaliŞir [2018]	?	?	?	?	+	+	**?**
McDonald [2019]	?	?	?	?	+	+	**?**
Morales-Muñoz [2017]	?	?	?	?	+	+	**?**
Ngoma [2010]	−	?	?	+	+	+	**−**
Randers [2021]	+	?	?	?	+	+	**?**
Saleem [2013]	+	?	?	?	+	+	**?**
Service [2021]	?	?	?	+	+	+	**?**
Shi [2019]	?	?	?	?	+	+	**?**
Tang [2019]	?	?	?	?	+	+	**?**
Vignapiano [2019]	+	?	−	?	+	+	**−**
Wang [2016]	?	?	−	?	+	+	**−**
Wu [2016]	?	?	−	?	+	+	**−**
Xiao [2017]	?	?	−	?	+	+	**−**
Xiu [2018]	?	?	?	?	+	+	**?**
Yang [2016]	+	+	?	?	+	+	**?**
Yang [2019]	?	+	?	?	+	+	**?**
Yang [2020]	+	?	?	?	+	+	**?**
Zhang [2013]	+	?	?	?	+	+	**?**
Zhang [2018]	+	?	?	?	+	+	**?**
Zhao [2019]	+	?	?	+	+	+	**?**
Zhou, Tang et al. [2019]	?	?	?	?	+	+	**?**
Zhou, Yu et al. [2019]	?	?	?	?	+	+	**?**
Zhou [2021]	?	?	?	?	+	+	**?**
Zong [2021]	+	?	?	?	+	+	**?**

**Table 2 brainsci-13-00299-t002:** Assignment of tests to neurological function. Neuropsychological tests: ANT: animal naming test, AVLT: auditory verbal learning Test, BACS: brief assessment of cognition in schizophrenia, BVMT-R: brief visuospatial memory test-revised, CANTAB: Cambridge neuropsychological test automated battery, CFT: category fluency test, CFET: Chinese facial emotion test, CTT-1/2: color trails test 1/2, COWAT: controlled oral word association test, CPT: continuous performance test, CPT-IP: continuous performance test-identical pairs, CPT-OX: flanked continuous performance test, CVLT: California verbal learning test, DART: Danish adult reading test, DSST: digit symbol substitution test, DVT: digit vigilance test, FAB: frontal assessment battery, FFT: finger tapping test, FOT: finger oscillation test, HVTL-R: Hopkins verbal learning test-revised, LNS: letter–number sequencing, LPS-3: Leistungsprüfsystem 3, MCCB: MATRICS Consensus Cognitive Battery, MSCEIT: Mayer–Salovey–Caruso emotional intelligence test, MWT: Mehrfachwortschatztest, NAB: neuro-psychological assessment battery, PASAT: paced auditory serial addition task, PennCNB: University of Pennsylvania computerized neurological battery, PGI: P.G.I. battery of brain dysfunction, RAVLT: Rey auditory verbal learning test, RBANS: repeatable battery for the assessment of neurocognitive status, ROCFT: Rey Osterrieth complex figure test, SOPT: self-ordered pointing task, SPT: spatial processing test, SCWT: Stroop color–word test, ToH: Tower of Hanoi, ToL: Tower of London, TAP: Testbatterie zur Aufmerksamkeitsprüfung, TASIT: the awareness of social inference test, TAVEC: Complutense verbal learning test, TMT-A/B: trail making test-A/B, VFT: verbal fluency test, VPA: verbal paired associates, WAIS: Wechsler adult intelligence scale, WAIS-III: Wechsler adult intelligence scale-III, WAIS-RC: Wechsler adult intelligence scale-Chinese revision, WCST: Wisconsin card sorting test, 15WoR: 15 words of Rey, WMS: Wechsler memory scale, WMS-III: Wechsler memory scale-III, WTAR: Wechsler test of adult reading.

Cognitive Domain	Cognitive Subdomain	Tests
Attention and Vigilance		CANTAB: rapid visual information processing; CPT; CPT-IP; CTT-1; D2; digit span-forward; DVT; PennCNB: continuous performance test; RBANS: digit span; WMS-III: spatial span-forward
Executive Function	Abstraction	PennCNB: conditional exclusion test; semantic similarities test; WAIS-III: similarities
	Cognitive Flexibility	CANTAB: intra/extra dimensional set shift; CTT-2; design fluency; FAB; TMT-B, WCST/WCST64
	Inhibition	SCWT: color–word subtest; TAP: go/no-go
	Planning	CANTAB: one touch stockings of Cambridge, stockings of Cambridge; ToH; ToL
	Reasoning and Problem Solving	Block design test; block diagram test; lPs:3, NAB: mazes; PennCNB: matrix reasoning test; spatial processing (block design); WAIS/WAIS-III/WAIS-RC: block design; WAIS-III: matrix reasoning, picture arrangement
Language	Comprehension	WAIS/WAIS-III: comprehension
	Naming	RBANS: picture naming
	Verbal Fluency	action (verb) fluency; ANT; BACS: verbal fluency; CFT: animal naming; COWAT; phonemic fluency test; RBANS: semantic fluency; semantic fluency test; verbal fluency; VFT
	Vocabulary	DART; MWT; vocabulary test; WAIS-III: vocabulary; WTAR
Social Cognition		Animated triangles task; CANTAB: emotion recognition task; CFET; faux pas recognition test; hinting task; MSCEIT; PennCNB: emotion differentiation test, emotion recognition test; emotion discrimination task; TASIT
Verbal Cognition	Verbal Learning	BACS: list learning; CVLT; HVLT-R; RAVLT/15WoR/AVLT; RBANS: list learning; TAVEC; VPA; WMS/WMS-III: logical memory/narrative memory
	Verbal Memory	Babcock story recall test; CERAD: WLM, word list recall; PennCNB: word memory test; RBANS: list recall, list recognition, story memory, story recall
Visuomotor Processing		BACS: token motor task, symbol coding; CANTAB: reaction time; DSST/WAIS: digit symbol/WAIS-III: digit symbol coding; FFT/FOT; grooved pegboard; PennCNB: finger tapping test; motor praxis test/mouse practice task; RBANS: coding; SCWT: color–word subtests; TMT-A; WAIS-III: symbol search
Visuo-spatial Cognition	Visual Analysis and Construction	PennCNB: line orientation; RBANS: figure copy, line orientation; ROCFT
	Visual Learning and Memory	BVMT-R; CANTAB: delayed matching to sample; paired associates learning; pattern recognition memory, spatial recognition memory; PennCNB: face memory; visual object learning test; PGI-visual recognition; RBANS: figure recall
Working Memory		BACS: digit sequencing; CANTAB: spatial span, spatial working memory; CPT-OX; digit span-backward; LNS; PASAT/PASAT-50; PennCNB: letter n-back task; serial subtraction; SOPT; TAP: 2-back task; verbal n-back task; visual n-back task; WAIS/WAIS-III: arithmetic; digit span-backward; WMS-III: spatial span-backward

**Table 3 brainsci-13-00299-t003:** Study characteristics and main results. Study populations: CHS: chronic schizophrenia, CHS-D: deficit chronic schizophrenia as defined by stable negative symptoms, CHS-ND: non-deficit chronic schizophrenia, CHS-NMR: chronic schizophrenia without motor retardation, CHS-NR: chronic schizophrenia treatment non-responders, CHS-MR: chronic schizophrenia with motor retardation, CHS-NOS: chronic schizophrenia without oxidative stress, CHS-OS: chronic schizophrenia with oxidative stress, CHS-PR: chronic schizophrenia treatment partial responders, HC: heathy controls, HRP: high risk of psychotic disorder, HRP-early: early stages of high risk for psychotic disorder, HRP-late: late stages of high risk of psychotic disorder, FEP: first-episode psychotic disorder, S-FEP: siblings of first-episode patients, FES: first-episode schizophrenia, FES-NR: non-remitted first-episode schizophrenia, FES-R: remitted first-episode schizophrenia, SZ: schizophrenic patients. Neuropsychological tests: ANT: animal naming test, AVLT: auditory verbal learning test, BACS: brief assessment of cognition in schizophrenia, BVMT-R: brief visuospatial memory test-revised, CANTAB: Cambridge neuropsychological test automated battery, CFT: category fluency test, CFET: Chinese facial emotion test, CTT-1/2: color trails test ½, COWAT: controlled oral word association test, CPT: continuous performance test, CPT-IP: continuous performance test-identical pairs, CPT-OX: flanked continuous performance test, CVLT: California verbal learning test, DART: Danish adult reading test, DSST: digit symbol substitution test, DVT: digit vigilance test, FAB: frontal assessment battery, FFT: finger tapping test, FOT: finger oscillation test, HVTL-R: Hopkins verbal learning test-revised, LNS: letter–number sequencing, LPS-3: Leistungsprüfsystem 3, MCCB: MATRICS Consensus Cognitive Battery, MSCEIT: Mayer–Salovey–Caruso emotional intelligence test, MWT: Mehrfachwortschatztest, PASAT: paced auditory serial addition task, PennCNB: University of Pennsylvania computerized neurological battery, PGI: P.G.I. battery of brain dysfunction, RAVLT: Rey auditory verbal learning test, RBANS: repeatable battery for the assessment of neurocognitive status, ROCFT: Rey Osterrieth complex figure test, SOPT: self-ordered pointing task, SCWT: Stroop color–word test, ToH: Tower of Hanoi, ToL: Tower of London, TAP: Testbatterie zur Aufmerksamkeitsprüfung, TASIT: the awareness of social inference test, TAVEC: Complutense verbal learning test, TMT-A/B: trail making test-A/B, VFT: verbal fluency test, VPA: verbal paired associates, WAIS: Wechsler adult intelligence scale, WAIS-III: Wechsler adult intelligence scale-III, WAIS-RC: Wechsler adult intelligence scale-Chinese revision, WCST: Wisconsin card sorting test, 15WoR: 15 words of Rey, WNS: Wechsler memory scale, WMS-III: Wechsler memory scale-III, WTAR: Wechsler test of adult reading.

Author	Patient Groups (n)	Control Groups (n)	Cognitive Tests (Cognitive Domains)	Main Results
Al-Dujaili **[26]**	CHS-NR (60) CHS-PR (55)	HC (43)	BACS (planning, verbal fluency, verbal learning, visuomotor processing, working memory)	CHS-NR < CHS-PR < HC for planning, verbal fluency, verbal learning, visuomotor processing, working memory.
Al-Hakeim **[27]**	CHS (120)	HC (54)	BACS (planning, verbal fluency, verbal learning, visuomotor processing, working memory)	CHS < HC for planning, verbal fluency, verbal learning, visuomotor processing, working memory.
Beck **[28]**	CHS (66)	HC (67)	BACS (planning, verbal fluency, verbal learning, visuomotor processing, working memory); WAIS-III: block design (reasoning and problem solving), vocabulary (vocabulary)	CHS < HC for reasoning and problem solving, vocabulary, working memory. When controlled for nationality, CHS < HC for verbal fluency, verbal learning, visuomotor processing (token motor task). When controlled for nationality and the interaction between nationality and group, CHS < HC for visuomotor processing (symbol coding). When controlled for nationality and the interaction between nationality and group, CHS = HC for planning.
Bliksted **[29]**	FES (36)	HC (36)	Animated triangles task (social cognition); BACS (planning, verbal fluency, verbal learning, visuomotor processing, working memory); DART (vocabulary); hinting task (social cognition); TASIT (social cognition); WAIS-III: matrix reasoning (reasoning and problem solving), block design (reasoning and problem solving), vocabulary (vocabulary), similarities (abstraction)	FEP < HC for abstraction, reasoning and problem solving, social cognition, verbal fluency, verbal learning, visuomotor processing, vocabulary, working memory.
Bosnjak Kuharic **[30]**	FEP (129)	HC (100)	Block design test (reasoning and problem solving); digit span-forward (attention and vigilance), Backward (working memory); DSST (visuomotor processing); FAB (cognitive flexibility); phonemic fluency test (verbal fluency); RAVLT (verbal learning); ROCFT (visual analysis and construction); SCWT (visuomotor processing, inhibition); semantic fluency test (verbal fluency); TMT-A (visuomotor processing) TMT-B (cognitive flexibility); WMS-III: VPA (verbal learning)	FEP < HC for cognitive flexibility, inhibition, reasoning and problem solving, verbal fluency, verbal learning, visual analysis and construction, visuomotor processing, working memory. FEP = HC for attention and vigilance.
Chattopadhyay **[31]**	CHS (34)	HC (47)	ANT (verbal fluency); design fluency; (cognitive flexibility) digit span-forward (attention and vigilance), backward (working memory); DSST (visuomotor processing); PGI-visual recognition (visual learning and memory); serial subtraction (working memory); verbal N -back test (working memory); visual N -back test (working memory); VPA (verbal learning)	CHS < HC for cognitive flexibility, verbal fluency, verbal learning, visual learning and memory, visuomotor processing, working memory. CHS = HC for attention and vigilance.
Chen **[32]**	FES (42)	HC (36)	MCCB (attention and vigilance, reasoning and problem solving, social cognition, verbal fluency, verbal learning, visual learning and memory, visuomotor processing, working memory)	FEP < HC for attention and vigilance, reasoning and problem solving, social cognition, verbal fluency, verbal learning, visual learning and memory, visuomotor processing, working memory.
Chen **[33]**	FES (50) CHS (158)	HC (40)	MCCB (attention and vigilance, reasoning and problem solving, social cognition, verbal fluency, verbal learning, visual learning and memory, visuomotor processing, working memory)	FES < CHS < HC for MCCB composite score and domains attention and vigilance, reasoning and problem solving, visual learning and memory, visuomotor processing (symbol coding), working memory. FES, CHS < HC for verbal fluency, verbal learning, visuomotor processing (TMT-A). FES < CHS, HC for social cognition.
Correa-Ghisays **[34]**	CHS (30)	HC (28)	FFT (visuomotor processing); ROCFT (visual analysis and construction); SCWT (visuomotor processing, inhibition); TAVEC: V1, V3, V4, V8, V10 (verbal learning); TMT-A (visuomotor processing) TMT-B (cognitive flexibility); VFT: semantic and phonemic forms; WCST (verbal fluency); WAIS-III: digit span-forward (attention and vigilance), backward (working memory), digit symbol coding (visuomotor processing), vocabulary (vocabulary)	CHS < HC for global cognitive score and cognitive domains attention and vigilance, cognitive flexibility, inhibition, verbal fluency, verbal learning, visual analysis and construction, visuomotor processing, working memory. CHS = HC for vocabulary.
Cuesta **[35]**	FEP (50)	HC (24) S-FEP (21)	BVMT-R (visual learning and memory); CPT-IP (attention and vigilance); MSCEIT (social cognition); TAVEC (verbal learning); TMT-A/B (visuomotor processing/cognitive flexibility); WAIS-III: digit span-forward/backward (attention and vigilance/working memory), digit symbol coding (visuomotor processing), LNS (working memory), symbol search (visuomotor processing), vocabulary (vocabulary); WCST-64 (cognitive flexibility); WMS-III: spatial span-forward/backward (attention and vigilance/working memory)	FEP < HC for visual learning and memory, vocabulary. FEP < S-FEP < HC for global cognition score and domains attention and vigilance, social cognition. FEP < S-FEP, HC for cognitive flexibility, verbal learning, visuomotor processing, working memory.
Da Motta **[36]**	CHS (38)	HC (97)	PennCNB (abstraction, attention and vigilance, reasoning and problem solving, social cognition, verbal memory, visual analysis and construction, visuomotor processing, working memory)	CHS < HC for abstraction, attention and vigilance, reasoning and problem solving, social cognition, verbal memory, visual learning and memory, visual analysis and construction, visuomotor processing, working memory.
De la Torre **[37]**	CHS (97)	HC (35)	RBANS-form A (attention and vigilance, naming, verbal fluency, verbal learning, verbal memory, visual analysis and construction, visual learning and memory, visuomotor processing)	CHS < HC for RBANS total performance and domains verbal fluency, verbal learning, verbal memory (list recognition, story memory, story recall), visual analysis and construction, visual learning and memory, visuomotor processing. CHS = HC for attention and vigilance, naming, verbal memory (list recall).
Eisenacher **[38]**	HRP (38)	HC (38)	MCCB (attention and vigilance, reasoning and problem solving, social cognition, verbal fluency, verbal learning, visual learning and memory, visuomotor processing, working memory); WCST (cognitive flexibility)	HRP < HC for MCCB composite score and domains reasoning and problem solving, verbal fluency, verbal learning, visuomotor processing, working memory. HRP = HC for attention and vigilance, cognitive flexibility, visual learning and memory, social cognition.
Ferretjans **[39]**	CHS (69)	HC (43)	BACS (planning, verbal fluency, verbal learning, visuomotor processing, working memory)	CHS < HC for BACS composite score and domains planning, verbal fluency, verbal learning, visuomotor processing, working memory.
Frommann **[40]**	HRP-early (116) HRP-late (89)	HC (87)	CPT-IP (attention and vigilance); LNS (working memory); MWT (vocabulary); AVLT (verbal learning); SOPT (working memory); TMT-A/B (visuomotor processing/cognitive flexibility); verbal fluency; (verbal fluency) WAIS-III: digit symbol coding (visuomotor processing)	HRP-late < HRP-early < HC for general cognitive score and domains cognitive flexibility, verbal learning, visuomotor processing (TMT A). HRP-late < HRP-early, HC for visuomotor processing (Digit Symbol Coding), working memory (SOPT). HRP-late, HRP-early < HC for verbal fluency. HRP-late < HC for attention and vigilance. HRP-late = HRP-early = HC for vocabulary, working memory (LNS).
Giordano **[41]**	CHS (114)	HC (63)	MCCB (attention and vigilance, reasoning and problem solving, social cognition, verbal fluency, verbal learning, visual learning and memory, visuomotor processing, working memory)	CHS < HC for attention and vigilance, reasoning and problem solving, social cognition, verbal fluency, verbal learning, visual learning and memory, visuomotor processing, working memory.
Guo **[42]**	FES (51)	HC (41)	BACS: symbol coding (visuomotor processing); BVMT-R (visual analysis and construction); CFT: animal naming (verbal fluency); HVLT-R (verbal learning); SCWT (visuomotor processing, inhibition); TMT-A (visuomotor processing); WMS-III: spatial span-forward/backward (attention and vigilance/working memory)	FEP < HC for attention and vigilance, inhibition, verbal fluency, verbal learning, visual learning and memory, visuomotor processing, working memory.
Hájková **[43]**	FEP (53)	HC (49)	CPT (attention and vigilance); RAVLT (verbal learning); ROCFT (visual analysis and construction); SCWT (visuomotor processing, inhibition); TMT-A/B (visuomotor processing/cognitive flexibility); verbal fluency (verbal fluency); WAIS-III: comprehension (comprehension), digit span-forward/backward (attention and vigilance/working memory), digit symbol coding (visuomotor processing), LNS (working memory), picture arrangement (reasoning and problem solving), similarities (abstraction), ToL (planning); WMS-III: logical memory (verbal learning), spatial span-forward/backward (attention and vigilance/working memory)	FES < HC for abstraction, attention and vigilance, cognitive flexibility, comprehension, inhibition, planning, reasoning and problem solving, verbal fluency, verbal learning, visual analysis and construction, visuomotor processing, working memory.
He **[44]**	FEP (115)	HC (113)	CANTAB: pattern recognition memory (visual learning and memory), rapid visual information processing (attention and vigilance); TMT-A/B (visuomotor processing/cognitive flexibility); WAIS: digit symbol test (visuomotor processing); WMS: logical memory (verbal learning)	FEP < HC for attention and vigilance, cognitive flexibility, verbal learning, visual learning and memory, visuomotor processing.
Konstantakopoulos **[45]**	CHS (54)	HC (53)	Babcock story recall test (verbal memory); Faux pas recognition test (social cognition); SCWT (visuomotor processing, inhibition); TMT-A/B (visuomotor processing/cognitive flexibility); WAIS: block design (reasoning and problem solving), digit span-backward (working memory), vocabulary (vocabulary); WCST-64 (cognitive flexibility)	CHS < HC for cognitive flexibility, inhibition, reasoning and problem solving, social cognition, verbal memory, visuomotor processing, vocabulary, working memory.
Koshiyama **[46]**	CHS (428)	HC (283)	CVLT (verbal learning); LNS (working memory); WCST (cognitive flexibility)	CHS < HC for cognitive flexibility, verbal learning, working memory.
Li **[47]**	HRP (34)	HC (37)	BACS: symbol coding (visuomotor processing); BVMT-R (visual learning and memory); CPT-IP (attention and vigilance); HVTL-R (verbal learning); SCWT (visuomotor processing, inhibition); TMT-A/B (visuomotor processing/cognitive flexibility)	HRP < HC for attention and vigilance, cognitive flexibility, inhibition, verbal learning, visual learning and memory, visuomotor processing (symbol coding, SCWT color–word subtest). HRP = HC for visuomotor processing (TMT-A).
Liu **[48]**	HRP (73) FES (44) CHS (34)	HC (72)	MCCB (attention and vigilance, reasoning and problem solving, social cognition, verbal fluency, verbal learning, visual learning and memory, visuomotor processing, working memory)	HRP, FES, CHS < HC for MCCB composite score and domains reasoning and problem solving, social cognition, verbal fluency, visual learning and memory, visuomotor processing. FES, CHS < HC for attention and vigilance. CHS < HRP, HC for attention and vigilance. CHS < HC for working memory. FES, HRP = HC for working memory. HRP, FES, CHS = HC for verbal learning.
Liu **[49]**	FES (31)	HC (33)	Block diagram test (reasoning and problem solving); MSCEIT: Managing Emotions (social cognition); RBANS (attention and vigilance, naming, verbal fluency, verbal learning, verbal memory, visual analysis and construction, visual learning and memory, visuomotor processing); TMT-A (visuomotor processing); vocabulary test (vocabulary)	FES < HC for RBANS composite score and domains attention and vigilance, naming, reasoning and problem solving, verbal fluency, verbal learning, verbal memory (immediate, delayed), visual analysis and construction, visual learning and memory, visuomotor processing. FES = HC for social cognition, vocabulary.
Maes, Sirivichayakul, Kanchanatawan et al. **[50]**	CHS-MR (40) CHS-NMR (39)	HC (40)	CANTAB: one touch stockings of Cambridge (planning), rapid visual information processing (attention and vigilance), spatial working memory (working memory); CERAD: VFT (verbal fluency), WLM (verbal memory), word list recall (verbal memory)	CHS-MR < CHS-NMR < HC for attention and vigilance, verbal fluency, verbal memory. CHS-MR, CHS-NMR < HC for planning, working memory.
Maes, Sirivichayakul, Matsumoto et al. **[51]**	CHS-NOS (39) CHS-OS (40)	HC (40)	CANTAB: one touch stockings of Cambridge (planning), spatial working memory (working memory); CERAD: VFT (verbal fluency), WLM (verbal memory), word list recall (verbal memory)	CHS-OS < CHS < HC for verbal fluency, verbal memory. CHS-OS, CHS < HC for planning, working memory.
Maes **[52]**	CHS-ND (40) CHS-D (40)	HC (40)	CANTAB: emotion recognition test (social cognition), intra-extra-dimensional set shift (cognitive flexibility), one touch stockings of Cambridge (planning), paired association learning (visual learning and memory), rapid visual information processing (attention and vigilance), spatial working memory (working memory); CERAD: VFT (verbal fluency), WLM (verbal memory), word list recall (verbal memory)	CHS-D < CHS-ND, HC for attention and vigilance (rapid visual information processing–detection), verbal memory (CERAD: word list recall), visual learning and memory CHS-D < CHS-ND < HC for social cognition, verbal fluency, verbal memory (CERAD: WLM). CHS-D, CHS-ND < HC for attention and vigilance (rapid visual information processing–speed), planning, working memory.– CHS-D < HC for cognitive flexibility.
Mançe ÇaliŞir **[53]**	CHS (17)	HC (23)	RAVLT: delayed recall, immediate memory, learning (verbal learning); WAIS: arithmetic (working memory), block design (reasoning and problem solving), comprehension (comprehension), digit span-forward/backward (attention and vigilance/working memory), digit symbol (visuomotor processing), similarities (abstraction); WCST (cognitive flexibility)	CHS < HC for cognitive flexibility, comprehension, visuomotor processing. CHS = HC for abstraction, attention and vigilance, reasoning and problem solving, verbal learning, working memory.
McDonald **[54]**	HRP (101)	HC (38)	BACS (planning, verbal fluency, verbal learning, visuomotor processing, working memory)	HRP < HC for visuomotor processing (token motor task). HRP = HC for planning, verbal fluency, verbal learning, visuomotor processing (symbol coding) working memory.
Morales-Muñoz **[55]**	FEP (38)	HC (38)	MCCB (attention and vigilance, reasoning and problem solving, social cognition, verbal fluency, verbal learning, visual learning and memory, visuomotor processing, working memory)	FEP < HC for attention and vigilance, reasoning and problem solving, social cognition, verbal fluency, verbal learning, visual learning and memory, visuomotor processing, working memory.
Ngoma **[56]**	FEP (188)	HC (153)	15WoR (verbal learning); COWAT (verbal fluency); d2 (attention and vigilance); FOT (visuomotor processing); LNS (working memory); ROCFT (visual analysis and construction); SCWT (visuomotor processing/inhibition); TMT-A/B (visuomotor processing/cognitive flexibility); WCST (cognitive flexibility)	CHS < HC for attention and vigilance, cognitive flexibility, inhibition, verbal fluency, verbal learning, visual analysis and construction, visuomotor processing, working memory.
Randers **[57]**	HRP (50)	HC (50)	BACS: digit sequencing (working memory), list learning (verbal learning), symbol coding (visuomotor processing), token motor task (visuomotor processing), verbal fluency (verbal fluency); CANTAB: delayed matching to sample (visual learning and memory), intra/extra-dimensional set shift (cognitive flexibility); rapid visual information processing (attention and vigilance), reaction time (visuomotor processing); spatial span-forward/backward (attention and vigilance/working memory), spatial working memory (working memory), stockings of Cambridge (planning); DART (vocabulary); TMT-A/B (visuomotor processing); WAIS-III: block design (reasoning and problem solving), matrix reasoning (reasoning and problem solving), similarities (abstraction); vocabulary (vocabulary)	HRP < HC for abstraction, attention and vigilance, cognitive flexibility, planning, reasoning and problem solving, verbal fluency, verbal learning, visual learning and memory, visuomotor processing, vocabulary, working memory.
Saleem **[58]**	FEP (20)	HC (15)	CANTAB: intra/extra dimensional set shift (cognitive flexibility), pattern recognition memory (visual learning and memory), reaction time (visuomotor processing), spatial recognition memory (visual learning and memory), stockings of Cambridge (planning); WTAR (vocabulary)	FEP < HC for cognitive flexibility, planning, visual learning and memory, visuomotor processing. FEP = HC for vocabulary.
Service **[59]**	CHS (160)	HC (717)	PennCNB: CPT (attention and vigilance), digit symbol test (visuomotor processing), emotion differentiation test (social cognition), emotion recognition test (social cognition), face memory test (visual learning and memory), letter N -back task (working memory), matrix reasoning test (reasoning and problem solving), motor praxis test (visuomotor processing)	CHS < HC for attention and vigilance, reasoning and problem solving, social cognition, visual learning and memory, visuomotor processing, working memory.
Shi **[60]**	CHS (230)	HC (656)	Action (verb) fluency (verbal fluency); color trails test-I/II (attention and vigilance/cognitive flexibility); grooved pegboard (visuomotor processing); MCCB (attention and vigilance, reasoning and problem solving, social cognition, verbal fluency, verbal learning, visual learning and memory, visuomotor processing, working memory); PASAT-50 (working memory); SCWT (visuomotor processing); WCST-64 (cognitive flexibility)	CHS < HC for attention and vigilance, cognitive flexibility, inhibition, reasoning and problem solving, social cognition, verbal fluency, verbal learning, visual learning and memory, visuomotor processing, working memory.
Tang **[61]**	CHS-D (51) CHS-ND (58)	HC (40)	DVT (attention and vigilance); PASAT (working memory); SCWT (visuomotor processing, inhibition); VFT: actions, animals (verbal fluency); WAIS-RC: block design (reasoning and problem solving)	CHS-D < CHS-ND < HC for inhibition, reasoning and problem solving, verbal fluency, visuomotor processing, working memory. CHS-D < CHS-ND, HC for attention and vigilance.
Vignapiano **[62]**	CHS (145)	HC (69)	MCCB (attention and vigilance, reasoning and problem solving, social cognition, verbal fluency, verbal learning, visual learning and memory, visuomotor processing, working memory)	CHS < HC for MCCB composite score and domains attention and vigilance, reasoning and problem solving, social cognition, verbal learning, visual learning and memory, visuomotor processing, working memory.
Wang **[63]**	FES (81)	HC (73)	BACS (planning, verbal fluency, verbal learning, visuomotor processing, working memory); COWAT (verbal fluency)	FEP < HC for BACS composite score and domains planning, verbal fluency, verbal learning, visuomotor processing, working memory.
Wu **[64]**	FES (79) CHS (132)	HC (124)	MCCB (attention and vigilance, reasoning and problem solving, social cognition, verbal fluency, verbal learning, visual learning and memory, visuomotor processing, working memory)	FES < HC for attention and vigilance, reasoning and problem solving, social cognition, verbal fluency, verbal learning, visual learning and memory, visuomotor processing, working memory. CHS < HC for attention and vigilance, reasoning and problem solving, social cognition, verbal fluency, verbal learning, visual learning and memory, visuomotor processing, working memory. CHS < FES for reasoning and problem solving, social cognition, verbal fluency, verbal learning, visuomotor processing (TMT-A), working memory (digital sequence). CHS = FES for attention and vigilance, visual learning and memory, visuomotor processing (symbol coding), working memory (spatial span).
Xiao **[65]**	FES (58)	HC (55)	Digit span-forward/backward (attention and vigilance); SCWT (visuomotor processing, inhibition); TMT-A/B (visuomotor processing, cognitive flexibility); VFT: animals, actions (verbal fluency)	FES < HC for cognitive flexibility, inhibition, verbal fluency, visuomotor processing, working memory. FES = HC for attention and vigilance.
Xiu **[66]**	FES (45) CHS (35)	HC (40)	RBANS-form A (attention and vigilance, naming, verbal fluency, verbal learning, verbal memory, visual analysis and construction, visual learning and memory, visuomotor processing)	CHS, FEP < HC for RBANS composite score and domains attention and vigilance, naming, verbal fluency, verbal learning, verbal memory (delayed), visual learning and memory, visuomotor processing. CHS < FEP < HC for verbal learning, verbal memory (immediate). CHS < HC for visual analysis and construction. FEP = HC for visual analysis and construction.
Yang **[67]**	FES (21) CHS (26) SZ (47) = FES (21) + CHS (26)	HC (45)	BACS: symbol coding (visuomotor processing); BVMT-R (visual learning and memory); CPT-IP (attention and vigilance); HVLT-R (verbal learning), SCWT (visuomotor processing, inhibition); TMT-A (visuomotor processing); WMS-III: spatial span-forward/backward (attention and vigilance/working memory)	SZ < HC for attention and vigilance, inhibition, verbal learning, visual learning and memory, visuomotor processing, working memory. When controlled for age, gender, years of education and body mass index, SZ = HC for visual learning and memory. CHS = FES for verbal learning, visual learning, visual memory. CHS < FES for attention and vigilance, inhibition, visuomotor processing, working memory. When controlled for age, gender, years of education body mass index and PANSS scores, CHS < FES for attention and vigilance, inhibition, visuomotor processing (SCWT color–word), working memory. When controlled for age, gender, years of education, body mass index and PANSS scores, CHS = FES for verbal learning, visual learning and memory, visuomotor processing (symbol coding, TMT-A).
Yang **[68]**	FES (34) CHS (31)	HC (35)	MCCB (attention and vigilance, reasoning and problem solving, social cognition, verbal fluency, verbal learning, visual learning and memory, visuomotor processing, working memory)	CHS, FES < HC for MCCB composite score and domains attention and vigilance, reasoning and problem solving, social cognition, verbal fluency, verbal learning, visual learning and memory, visuomotor processing, working memory.
Yang **[69]**	CHS (32)	HC (30)	BACS: symbol coding (visuomotor processing); BVMT-R (visual learning and memory); CPT-IP (attention and vigilance); HVLT-R (verbal learning); SCWT (visuomotor processing, inhibition); TMT-A (visuomotor processing); WMS-III: spatial span-forward/backward (attention and vigilance/working memory)	CHS < HC for attention and vigilance, inhibition, verbal learning, visual learning and memory, visuomotor processing, working memory. When controlled for age, gender, years of education and body mass index, CHS = HC for visual learning and memory.
Zhang **[70]**	FES (77)	HC (75)	RBANS (attention and vigilance, naming, verbal fluency, verbal learning, verbal memory, visual analysis and construction, visual learning and memory, visuomotor processing)	FES < HC for attention and vigilance, naming, verbal fluency, verbal learning, verbal memory. FES = HC for visual analysis and construction.
Zhang **[71]**	FES (32)	HC (29)	AVLT (verbal learning); digit span-forward/backward (attention and vigilance/working memory); SCWT (visuomotor processing, inhibition); semantic similarity test (abstraction); TMT-A/B (visuomotor processing, inhibition); VFT (verbal fluency); WCST (cognitive flexibility);	FES < HC for abstraction, attention and vigilance, cognitive flexibility, inhibition, verbal fluency, verbal learning, visuomotor processing, working memory.
Zhao **[72]**	FES-R (65) FES-NR (45)	HC (58)	CFET (social cognition); digit span-forward/backward (attention and vigilance/working memory); HVLT-R (verbal learning); TMT-A/B (visuomotor processing/cognitive flexibility); SCWT (visuomotor processing, inhibition); VFT: actions, animals (verbal fluency)	FES-NR < FES-R < HC for cognitive flexibility, inhibition, verbal fluency (animals), verbal learning, visuomotor processing. FES-NR < FES-R, HC for social cognition, verbal fluency (actions). FES-NR, FES-R = HC for attention and vigilance, working memory.
Zhou, Tang et al. **[73]**	CHS-D (33) CHS-ND (41)	HC (40)	ANT (verbal fluency); COWAT (verbal fluency); DVT (attention and vigilance); SCWT (visuomotor processing, inhibition); spatial processing block design (reasoning and problem solving); TMT-A/B (visuomotor processing/cognitive flexibility); WAIS-RC: block design (reasoning and problem solving)	CHS-D < CHS-ND, HC for attention and vigilance. CHS-D < CHS-ND < HC for cognitive flexibility, reasoning and problem solving (spatial processing: block design), verbal fluency, visuomotor processing CHS-D, CHS-ND < HC for inhibition, reasoning and problem solving (WAIS-RC: block design)
Zhou, Yu et al. **[74]**	CHS-D (37) CHS-ND (38)	HC (38)	ANT (verbal fluency); COWAT (verbal fluency); DVT (attention and vigilance); SCWT (visuomotor processing, inhibition); spatial processing block design (reasoning and problem solving); TMT-A/B (visuomotor processing/cognitive flexibility); WAIS-RC: block design (reasoning and problem solving)	CHS-D < CHS-ND < HC for attention and vigilance, reasoning and problem solving (spatial processing: block design), visuomotor processing (SCWT color–word subtests). CHS-D, CHS-ND < HC for cognitive flexibility, inhibition, reasoning and problem solving (WAIS-RC: block design), verbal fluency, visuomotor processing (TMT-A).
Zhou **[75]**	CHS-D (58) CHS-ND (93)	HC (113)	ANT (verbal fluency); COWAT (verbal fluency); DVT (attention and vigilance); SCWT (visuomotor processing, inhibition); spatial processing block design (reasoning and problem solving); TMT-A/B (visuomotor processing/cognitive flexibility); WAIS-RC: block design (reasoning and problem solving)	CHS-D < CHS-ND < HC for attention and vigilance, reasoning and problem solving (WAIS-RC: block design), verbal fluency (ANT), visuomotor processing (SCWT color–word subtests). CHS-D, CHS-ND < HC for cognitive flexibility, inhibition, reasoning and problem solving (spatial processing block design), verbal fluency (COWAT), visuomotor processing (TMT-A).
Zong **[76]**	FES (42)	HC (38)	Digit span-forward/backward (attention and vigilance/working memory); TMT-A/B (visuomotor processing/cognitive flexibility)	FES < HC for attention and vigilance, cognitive flexibility, visuomotor processing, working memory.

**Table 4 brainsci-13-00299-t004:** Summary of main results for first-episode stage and chronic stage patients. The included cognitive subdomains were each investigated by at least three studies. The number of studies that assessed a particular domain is presented in the brackets after each domain. Subdomains (N = number of studies per stage that investigated this domain): abstraction (N_FES_ = 3), attention and vigilance (N_FES_ = 20, N_CHS_ = 20), cognitive flexibility (N_FES_ = 10, N_CHS_ = 9), inhibition (N_FES_ = 8, N_CHS_ = 9), naming (N_FES_ = 3), planning (N_FES_ = 3, N_CHS_ = 7), reasoning and problem solving (N_FES_ = 10, N_CHS_ = 16), social cognition (N_FES_ = 9, N_CHS_ = 11), verbal fluency (N_FES_ = 18, N_CHS_ = 22), verbal learning (N_FES_ = 20, N_CHS_ = 19), verbal memory (N_FES_ = 3, N_CHS_ = 6), visual analysis and construction (N_FES_ = 6, N_CHS_ = 4), visual learning and memory (N_FES_ = 13, N_CHS_ = 14), visuomotor processing (N_FES_ = 22, N_CHS_ = 25), vocabulary (N_FES_ = 4, N_CHS_ = 3), working memory (N_FES_ = 18, N_CHS_ = 24). Abstraction (N_CHS_ = 2) and naming (N_CHS_ = 2) were not included for the chronic stage group due to lack of sufficient data, and comprehension (N_FES_ = 1, N_CHS_ = 1) was not included at all for the same reason. Calculation of quantiles: percentages were calculated per subgroup (first-episode or chronic stage patients) using the number of studies that reported deficits in a particular domain divided by the total number studies that assessed this domain. Since some of the studies included groups of both first-episode and chronic-stage patients, they have been counted multiple times for the results summary.

Patient Groups vs. Healthy Controls	First-Episode Stage N = 23 Studies	Chronic Stage N = 29 Studies
100% of studies reported deficits	Abstraction (N = 3), cognitive flexibility (N = 10), inhibition (N = 8), naming (N = 3), planning (N = 3), reasoning and problem solving (N = 10), verbal fluency (N = 18), verbal memory (N = 3), visuomotor processing (N = 22)	Cognitive flexibility (N = 9), inhibition (N = 9), verbal fluency (N = 22), verbal memory (N = 6), visual analysis and construction (N = 4), visuomotor processing (N = 25)
More than 75% of studies report deficits	Attention and vigilance (N = 20), social cognition (N = 9), verbal learning (N = 20), visual learning and memory (N = 13), working memory (N = 18)	Attention and vigilance (N = 20), planning (N = 7), reasoning and problem solving (N = 16), social cognition (N = 11), verbal learning (N = 19), visual learning and memory (N = 14), working memory (N = 24)
More than 50% of studies report deficits	Visual analysis and construction (N = 6)	Vocabulary (N = 3)
More than 25% of studies report deficits	Vocabulary (N = 4)	none
25% of studies or fewer report deficits	none	none

**Table 5 brainsci-13-00299-t005:** Results of N = 6 studies that included a direct comparison of patients diagnosed with first-episode stage (FES) and chronic stage (CHS) psychotic disorders, relative to healthy control (HC) groups. The following cognitive subdomains were included, all reported at least once across the six studies: attention and vigilance, inhibition, naming, reasoning and problem solving, social cognition, verbal fluency, verbal learning, verbal memory, visual analysis and construction, visual learning and memory, visuomotor processing, and working memory. Scoring: As some studies reported different results for subtests of the same domain (e.g., TMT-A vs. BACS: symbol coding) results were calculated based on subtests. All subtests were counted separately, even if several subtests were reported for the same domain within one study. Sums: Sums for each domain and result pattern were calculated and are shown in the right column and in the bottom row, respectively. These sums correspond to the number of tests across all six studies that reported results in the respective cognitive domains, as well as for the respective comparison of study groups.

	CHS < FES < HC (3 Studies)	FES < CHS < HC (1 Study)	(CHS = FES) < HC (6 Studies)	FES < (CHS = HC) (1 Study)	CHS < HC/ FES = HC (2 Studies)	CHS = FES = HC (2 Studies)	Sum of Significant Results per Domain
Attention and vigilance (6 studies)	2	1	4	0	0	0	7
Inhibition (1 study)	1	0	0	0	0	0	1
Naming (1 study)	0	0	1	0	0	0	1
Reasoning and problem solving (4 studies)	1	1	2	0	0	0	4
Social cognition (4 studies)	1	0	2	1	0	0	4
Verbal fluency (5 studies)	1	0	4	0	0	0	5
Verbal learning (6 studies)	2	0	3	0	0	1	6
Verbal memory (1 study)	1	0	3	0	0	0	4
Visual analysis and construction (1 study)	0	0	0	0	2	0	2
Visual learning and memory (5 studies)	0	1	3	0	0	1	5
Visuomotor processing (6 studies)	3	1	9	0	0	0	13
Working memory (5 studies)	2	2	3	0	2	0	9
Sum of significant tests per comparison	14	6	34	1	4	2	

**Table 6 brainsci-13-00299-t006:** Recommendations to lower the risk of bias in primary studies that focus on neurocognitive deficits in patients diagnosed with psychotic disorders.

Improve the matching of clinical and control groups based on premorbid (verbal) intelligence measurements of the patients, as well as age and gender. Control for moderating effects of medication on the correlation between psychotic symptom severity and cognitive deficits. Improve the characterization of clinical groups concerning the comorbid psychiatric or neurological illnesses that may impact cognitive performance. Perform specific analyses as well as data reporting for subgroups of patients diagnosed with psychotic disorders, e.g., schizophrenia spectrum disorders, persistent delusional disorders, or acute psychotic disorders. Execute blinding procedures of assessors and study groups concerning the characteristics and the number of study groups, as well as the hypotheses, to avoid an overestimation of differences between clinical and healthy groups. Apply a generally used cut-off when reporting clinically significant cognitive impairments (e.g., one standard deviation below the mean). Reduce the heterogeneity of cognitive outcome measures by using comparable, well-defined cognitive domains and respective psychometric tests across studies.

**Table 7 brainsci-13-00299-t007:** Recommendations for the clinical treatment of patients with psychotic disorders.

A neuropsychological assessment of cognitive domains evaluated in the current systematic review seems to be essential in advance of any psychotherapeutic intervention. Regular follow-up assessments are recommended in intervals of about 6–12 months throughout the treatment process. This is in line with recent advances towards the development of guidelines for the assessment of cognitive impairment in schizophrenia [96]. Specific cognitive behavioral interventions, such as a meta-cognitive training and cognitive remediation approaches, are recommended by the S3 guidelines on schizophrenia [97]. These need to be adjusted depending on the results of the neuropsychological assessment. We recommend a high level of awareness of severe cognitive impairment already at the stage of the first psychotic episode. A well-structured, focused, and supportive therapeutic environment needs to be provided with the aim of compensating for limitations in memory and executive functions, as reported in this systematic review, and to prevent patients from feelings of over-stimulation and frustration. Here, we recommend following the guidelines of adapting cognitive behavioral therapy interventions for patients with cognitive impairments [98,99]. The application of cognitive compensatory approaches [94] may help patients to maintain a high level of everyday functioning and perceived quality of life despite cognitive limitations. These strategies are, for example, the training of internal self-management strategies, as well as specific adjustments to the external environment, such as breaking down complex routines into chunks, and the use of external devices in support of memory functions.

## Data Availability

We embrace the values of openness and transparency in science (http://www.researchtransparency.org/, accessed on 24 October 2019). The pre-registration and open data are available at https://osf.io/nyk4s, accessed on 24 October 2019.

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
