# Peer review of "Neurocognitive Deficits in First-Episode and Chronic Psychotic Disorders: A Systematic Review from 2009 to 2022"

_brainsci, 2023, doi:10.3390/brainsci13020299_

Round 1
Reviewer 1 Report
Manuscript number: brainsci-2129953
Manuscript title: Neurocognitive Deficits in First-Episode and Chronic Psychotic Disorders: A Systematic Review
This is a delightful SR of studies on cognitive deficits in both early and chronic stages of schizophrenia spectrum disorders. Despite, I believe the paper is valuable, there are several issues, particularly relevant to the presentation, that needs to be reviewed to enhance the scientific merit of the paper.
- Is the only motivation for performing this SR the inclusion of the new studies that previous meta-analyses on first-episode and chronic psychotic disorders did not cover? Or is there any other novelty such as methodology or perspective in this review?
- The rationale for excluding schizoaffective disorder studies should be mentioned since it is a schizophrenia spectrum disorder or a chronic psychotic disorder.
- Mentioning treatment implications is a good idea in such an SR, however, I think the authors should give detail on how they formulated such recommendations presented in Table 7.
- The risk of bias is properly evaluated.
- Hypotheses are well defined.
Author Response
Reviewer 1:
Manuscript title: Neurocognitive Deficits in First-Episode and Chronic Psychotic Disorders: A Systematic Review
This is a delightful SR of studies on cognitive deficits in both early and chronic stages of schizophrenia spectrum disorders. Despite, I believe the paper is valuable, there are several issues, particularly relevant to the presentation, that needs to be reviewed to enhance the scientific merit of the paper.
Is the only motivation for performing this SR the inclusion of the new studies that previous meta-analyses on first-episode and chronic psychotic disorders did not cover? Or is there any other novelty such as methodology or perspective in this review?
Thank you for your comment. Beyond the inclusion of new studies from 2009 to 2022, we also performed a Cochrane Risk Assessment that provided a substantial quality control for the included studies. This has not been done by previous reviews and meta-analyses. The assessment included the domains “selection bias”, “clear diagnostics”, “blinding of patients”, “detection bias” (i.e., blinding of assessors), “complete vs. incomplete data reporting”, and “free of selective result reporting”. Based on these categories, we also provide six practical recommendations for the design of future primary studies to lower the risk of bias, and to improve the synthesis and replicability of research outcomes on neurocognitive deficits in psychotic disorders (Table 6).
In contrast to previous reviews and meta-analyses on cognitive impairment in psychotic disorders, we also provide a clear assignment of cognitive tests to specific cognitive functions across the included studies, thus enhancing the interpretability of research outcomes (Tables 2 and 3).
Furthermore, based on our specific outcomes concerning the degree of cognitive impairment at different stages of psychotic disorders, we provide recommendations for a stronger incorporation of neurocognitive performance factors in the planning and execution of clinical treatments for patients diagnosed with psychotic disorders (Table 7).
Our methodological advances concerning the quality control of included studies, our specification of applied tests regarding the measured cognitive functions, as well as our outcome-driven specific recommendations for the clinical practice are novel and of substantial value for the research community.
The rationale for excluding schizoaffective disorder studies should be mentioned since it is a schizophrenia spectrum disorder or a chronic psychotic disorder.
We have excluded schizoaffective disorders since there are substantial differences concerning the genetic origins, the symptoms, as wells as the course of illness relative to other schizophrenia spectrum disorders [1]. There have been also conflicting results concerning the level of cognitive impairment of patients diagnosed with schizoaffective disorder [2]. Hence, we belief that schizoaffective disorder should be analysed as a separate category in systematic reviews and meta-analyses on cognitive impairment in psychotic disorders. It would be a valuable project to extend our systematic review by including and comparing our findings with those of illnesses from a broader psychotic spectrum.
Mentioning treatment implications is a good idea in such an SR, however, I think the authors should give detail on how they formulated such recommendations presented in Table 7.
Many thanks for this comment. We have now included further references of already published guidelines that support our recommendations presented in Table 7:
|
1. A neuropsychological assessment of cognitive domains evaluated in the current systematic review seems to be essential in advance of any psychotherapeutic intervention. Regular follow-up assessments are recommended in intervals of about 6-12 months throughout the treatment process. This is in line with recent advances towards the development of guidelines for the assessment of cognitive impairment in schizophrenia [3]. 2. Specific cognitive-behavioral interventions, such as a meta-cognitive training and cognitive remediation approaches, are recommended by the S3 guidelines on schizophrenia [4]. Those need to be adjusted depending on results of the neuropsychological assessment. We recommend a high level of awareness of severe cognitive impairment already at the stage of the first psychotic episode. 3. A well-structured, focused, and supportive therapeutic environment needs to be provided with the aim to compensate for limitations in memory and executive functions, as reported in this systematic review, and to prevent patients from feelings of over-stimulation and frustration. We here recommend following the guidelines of adapting cognitive behavioral therapy interventions for patients with cognitive impairment [5,6]. 4. The application of cognitive compensatory approaches [7] may help patients to maintain a high level of everyday functioning and perceived quality of life despite cognitive limitations. Those strategies are, for example, the training of internal self-management strategies, as well as specific adjustments to the external environment, such as the down-breaking of complex routines into chunks, and the use of external devices in support of memory functions. |
Table 7. Recommendations for the clinical treatment of patients with psychotic disorders.
The risk of bias is properly evaluated.
Hypotheses are well defined.
References:
- Cardno, A.G.; Owen, M.J. Genetic relationships between schizophrenia, bipolar disorder, and schizoaffective disorder. Schizophr Bull 2014, 40, 504-515, doi:10.1093/schbul/sbu016.
- Lynham, A.J.; Cleaver, S.L.; Jones, I.R.; Walters, J.T.R. A meta-analysis comparing cognitive function across the mood/psychosis diagnostic spectrum. Psychological Medicine 2022, 52, 323-331, doi:10.1017/S0033291720002020.
- Vita, A.; Gaebel, W.; Mucci, A.; Sachs, G.; Erfurth, A.; Barlati, S.; Zanca, F.; Giordano, G.M.; Birkedal Glenthøj, L.; Nordentoft, M.; et al. European Psychiatric Association guidance on assessment of cognitive impairment in schizophrenia. European Psychiatry 2022, 65, e58, doi:10.1192/j.eurpsy.2022.2316.
- Hasan, A.; Falkai, P.; Lehmann, I.; Janssen, B.; Wobrock, T.; Zielasek, J.; Gaebel, W. [Revised S3 guidelines on schizophrenia : Developmental process and selected recommendations]. Nervenarzt 2020, 91, 26-33, doi:10.1007/s00115-019-00813-y.
- Rossiter, R.; Holmes, S. Access all areas: creative adaptations for CBT with people with cognitive impairments – illustrations and issues. The Cognitive Behaviour Therapist 2013, 6, e9, doi:10.1017/S1754470X13000135.
- Gallagher, M.; McLeod, H.J.; McMillan, T.M. A systematic review of recommended modifications of CBT for people with cognitive impairments following brain injury. Neuropsychological Rehabilitation 2019, 29, 1-21, doi:10.1080/09602011.2016.1258367.
- Allott, K.; van-der-El, K.; Bryce, S.; Parrish, E.M.; McGurk, S.R.; Hetrick, S.; Bowie, C.R.; Kidd, S.; Hamilton, M.; Killackey, E.; et al. Compensatory Interventions for Cognitive Impairments in Psychosis: A Systematic Review and Meta-Analysis. Schizophrenia Bulletin 2020, 46, 869-883, doi:10.1093/schbul/sbz134.
Reviewer 2 Report
Thank you for giving me the opportunity to review this manuscript.
I think it is necessary to revise the manuscript.
1) Please change the title as "Neurocognitive Deficits in First-Episode and Chronic Psychotic Disorders: An Updated Systematic Review from 2009 to 2022".
2) Please attach the PRISMA 2020 checklist and fill in the page numbers.
3) Please describe why the authors selected Cochrane risk of bias tool in spite that this study is not a systematic review of randomized controlled trials. Why did authors avoid selecting ROBINS or GRADE to assess the risk of bias in non-randomized studies.
4) Please show one table to explain what kind of scales were used to assess each cognitive domains in each study.
5) Please avoid using those abbreviations because those are not used in clinical practice, I think. Please define and classify the type of psychotic disorders in the method section. Why was it necessary to divide chronic schizophrenia into chronic schizophrenia, deficit chronic schizophrenia as defined by stable negative symptoms, non-deficit chronic schizophrenia, chronic schizophrenia without motor retardation, chronic schizophrenia treatment-non-responders, chronic schizophrenia with motor retardation, chronic schizophrenia without oxidative stress, chronic schizophrenia with oxidative stress, chronic schizophrenia treatment-partial-responders. Please explain why those divisions were meaningful. Furthermore, please define and classify the type of early psychosis in the method section. Why was it necessary to divide early psychosis into high risk of psychotic disorder, early stages of high risk for psychotic disorder, late stages of high risk of psychotic disorder, first-episode psychotic disorder, siblings of first-episode patients, first-episode schizophrenia, non-remitted first-episode schizophrenia, remitted first-episode schizophrenia, schizophrenic patients. Please explain why these divisions were clinically meningful.
I think it is necessary to revise the manuscript.
Author Response
Reviewer 2:
Thank you for giving me the opportunity to review this manuscript.
I think it is necessary to revise the manuscript.
1) Please change the title as "Neurocognitive Deficits in First-Episode and Chronic Psychotic Disorders: An Updated Systematic Review from 2009 to 2022".
Thank you for this clarification. We have now changed the title of the manuscript according to your recommendation: “Neurocognitive Deficits in First-Episode and Chronic Psychotic Disorders: An Updated Systematic Review from 2009 to 2022”.
2) Please attach the PRISMA 2020 checklist and fill in the page numbers.
Many thanks for this comment. We have now attached the PRISMA 2020 checklist with the respective page numbers, as well as uploaded it on the OSF platform.
3) Please describe why the authors selected Cochrane risk of bias tool in spite that this study is not a systematic review of randomized controlled trials. Why did authors avoid selecting ROBINS or GRADE to assess the risk of bias in non-randomized studies.
Many thanks for suggesting alternative tools for assessing the risk of bias (ROBINS) and the quality of evidence (GRADE) of our systematic review.
Both ROBINS and the Cochrane risk of bias tool are optimized for intervention studies. For intervention studies there are some important differences between the Cochrane risk of bias tool for randomized controlled trials and the ROBINS tool for non-randomized trials. The Cochrane risk of bias tool additionally includes (1) the random sequence generation and (2) the allocation concealment. The ROBINS tool includes (1) whether baseline confounding occurs when one or more prognostic variables also predict the intervention received at baseline, (2) whether a bias is introduced by either differential or non-differential misclassification of intervention status, or (3) whether a bias is due to deviations from intended interventions. For the included intervention studies, we only assessed differences at the pre-intervention baseline assessment. Thus, we only included risk of bias domains of the Cochrane risk of bias tool that are also part of the ROBINS tool, i.e., selection bias, bias due to missing data, bias in measurement of outcomes, and bias in selection of the reported results. We now clarify the risk of bias domains that were relevant to our systematic review citing both tools (see page 6 under “Risk of bias assessment”).
The GRADE system may be applied to rate the (overall) quality of evidence of a particular outcome across studies and is mainly developed for the assessment of intervention studies. It includes the risk of bias, imprecision, inconsistency, indirectness, and publication bias. Since we did not conduct a meta-analysis, we cannot provide an estimate for the overall effect size. Thus, we cannot provide estimates for the imprecision (i.e., 95% confidence intervals), the inconsistency (i.e., measures of heterogeneity such as I2), and the publication bias (which requires complex statistical tools [1]). The domain “indirectness” applies to the assessment and evaluation of interventions, and thus is not relevant for our systematic review. In our rating of evidence, we address the risk of bias and the problem of heterogeneity on a qualitative level, which is suitable to our approach of a systematic review. A full rating according to the GRADE system, however, may rather apply to a systematic review or meta-analysis which assesses the evidence of interventions and may provide estimates for the other domains.
4) Please show one table to explain what kind of scales were used to assess each cognitive domains in each study.
Many thanks for this comment. We have now included a column in Table 3 showing the assignment of cognitive tests to specific cognitive functions for each individual study. A general assignment of cognitive tests to specific cognitive domains and subdomains is provided in Table 2.
5) Please avoid using those abbreviations because those are not used in clinical practice, I think. Please define and classify the type of psychotic disorders in the method section. Why was it necessary to divide chronic schizophrenia into chronic schizophrenia, deficit chronic schizophrenia as defined by stable negative symptoms, non-deficit chronic schizophrenia, chronic schizophrenia without motor retardation, chronic schizophrenia treatment-non-responders, chronic schizophrenia with motor retardation, chronic schizophrenia without oxidative stress, chronic schizophrenia with oxidative stress, chronic schizophrenia treatment-partial-responders. Please explain why those divisions were meaningful.
The list of subgroups in the results section on page 11 includes all study samples, as defined by the primary studies of this systematic review. This is not our categorization of psychotic disorders, but rather an overview of all subcategories used in previous primary studies. The respective abbreviations have been extracted from those previous primary studies as well. We have now included a sentence on page 12 to clarify our approach:
“For our specific hypotheses, we here focussed on the main categories “first-episode” and “chronic stage” of psychosis, under which all respective subgroups above, as defined by primary studies, were summarized. For exploratory reasons, we also included the categories “high-risk” and “siblings of first-episode patients.”
6) Furthermore, please define and classify the type of early psychosis in the method section. Why was it necessary to divide early psychosis into high risk of psychotic disorder, early stages of high risk for psychotic disorder, late stages of high risk of psychotic disorder, first-episode psychotic disorder, siblings of first-episode patients, first-episode schizophrenia, non-remitted first-episode schizophrenia, remitted first-episode schizophrenia, schizophrenic patients. Please explain why these divisions were clinically meaningful.
These are subgroups defined by the primary studies that we included in this systematic review. Under “synthesis of study outcomes” we now clarify our approach on page 8 in the methods section:
“Concerning our specific hypotheses on the cognitive impairment of patients with first-episode and chronic psychotic disorders, we extracted and summarized patient groups belonging to either one of those categories from primary studies. For exploratory reasons, we also extracted and summarized the data of study groups carrying a high-risk for psychosis and those of siblings of first-episode patients.”
I think it is necessary to revise the manuscript.
References:
- Woll, C.F.J.; Schönbrodt, F.D. A Series of Meta-Analytic Tests of the Efficacy of Long-Term Psychoanalytic Psychotherapy. European Psychologist 2019, 25, 51-72, doi:10.1027/1016-9040/a000385.

Reviewer 3 Report
The authors present a systematic review looking at studies that evaluated cognition in first-episode and chronically ill schizophrenia patients compared to healthy controls. The systematic review is done well. Here are a few comments for consideration:
-"All included studies had per-formed an assessment of neurocognitive functioning by using reliable, valid, and objective neuropsychological instruments as outcomes (O)." - how was this defined? Did the tool have to have a peer-reviewed publication establishing its validity? Maybe a slight expansion of detail here would help
-Consider finding a way to present Table 3 such as using check boxes for cognitive tests utilized in each study and only presenting the results (the > or < group comparisons) and then presenting the expanded results comments in a supplementary table. I think this information is important and done well but the table is large and hard to read (multi page) in the current format.
-Something that should be considered and commented on are the treatment characteristics for the chronically ill studies. Do these studies report the antipsychotic treatment regimens? CPZE? If so, are the heterogeneous across studies and what implications, if any, would antipsychotic treatment have on your stability hypothesis?
-
Author Response
Reviewer 3:
The authors present a systematic review looking at studies that evaluated cognition in first-episode and chronically ill schizophrenia patients compared to healthy controls. The systematic review is done well. Here are a few comments for consideration:
"All included studies had per-formed an assessment of neurocognitive functioning by using reliable, valid, and objective neuropsychological instruments as outcomes (O)." - how was this defined? Did the tool have to have a peer-reviewed publication establishing its validity? Maybe a slight expansion of detail here would help
Many thanks for this comment. Yes indeed, all included studies are peer-reviewed publications written in English and published from international, well-established journals. All included cognitive tests from those studies are standardized tests fulfilling the criteria of a reliable, valid, and objective neuropsychological assessment instrument.
We have now included the following sentence in the methods section under “inclusion criteria” on page 5:
“We included primary, peer-reviewed studies published in English by international, well-established journals.”
Consider finding a way to present Table 3 such as using check boxes for cognitive tests utilized in each study and only presenting the results (the > or < group comparisons) and then presenting the expanded results comments in a supplementary table. I think this information is important and done well but the table is large and hard to read (multi page) in the current format.
Many thanks for this helpful feedback. We have now reduced the size and complexity of Table 3 by moving the demographic variables to the supplementary material. We have also clarified the assignment of cognitive tests to specific cognitive functions in the “cognitive tests” column for each individual study, making the results in the last column of the table easier to interpret.
Something that should be considered and commented on are the treatment characteristics for the chronically ill studies. Do these studies report the antipsychotic treatment regimens? CPZE? If so, are the heterogeneous across studies and what implications, if any, would antipsychotic treatment have on your stability hypothesis?
This is an important remark. We have now quantified the number of studies on chronically ill patients with and without antipsychotic treatment. We now write in the results section on page 14:
“We also assessed the putative impact of medication on the degree of reported cognitive impairment across studies on patients at the chronic stage of illness. In 38 percent of those studies antipsychotic treatment was applied to chronically ill patients. 14 percent of studies reported a mixed sample of patients, in which the majority but not all patients received antipsychotic medication. 7 percent of studies reported that no medication was given to patients. Most studies, i.e. 41 percent, did not provide sufficient information on medication. However, across the studies that reported on medication, there was no difference concerning the degree of cognitive impairment in chronically ill patients depending on the antipsychotic treatment.”

Round 2
Reviewer 2 Report
Thank you for revising the manuscript.
I think this manuscript would be suitable for publication in this journal.